# Large-Area Ordered Palladium Nanostructures by Colloidal Lithography for Hydrogen Sensing

**DOI:** 10.3390/molecules27186100

**Published:** 2022-09-18

**Authors:** Feng Xu, Zhiliang Zhang, Jun Ma, Churong Ma, Bai-Ou Guan, Kai Chen

**Affiliations:** Guangdong Provincial Key Laboratory of Optical Fiber Sensing and Communications, Institute of Photonics Technology, Jinan University, Guangzhou 511443, China

**Keywords:** palladium, hydrogen sensing, colloidal lithography, nanohole, surface plasmon

## Abstract

Reliable gas sensors are very important for hydrogen (H_2_) gas detection and storage. Detection methods based on palladium (Pd) metal are cost-effective and widely studied. When Pd is exposed to H_2_, it turns into palladium hydride with modified optical properties, which thus can be monitored for H_2_ sensing. Here, we fabricated large-area Pd nanostructures, including Pd nanotriangles and nanohole arrays, using colloidal lithography and systematically studied their H_2_-sensing performance. After hydrogen absorption, both the Pd nanoholes and nanotriangles showed clear transmittance changes in the visible–near infrared range, consistent with numerical simulation results. The influences of the structural parameters (period of the array *P* and diameter of the nanohole *D*) of the two structures are further studied, as different structural parameters can affect the hydrogen detection effect of the two structures. The nanohole arrays exhibited bigger transmittance changes than the nanotriangle arrays.

## 1. Introduction

Hydrogen is a colorless, odorless and highly flammable gas. It can be produced from water and has huge potential applications as an alternative to fossil fuels. As a renewable fuel, hydrogen has the potential to mitigate the global warming problem associated with fossil fuel consumption, because no carbon emissions are produced when hydrogen is consumed [1,2,3]. Hydrogen is also an important industrial raw material, widely used in petroleum, electronics, metallurgy, the aerospace industry and many other fields. However, H_2_ has a high diffusion coefficient (0.16 cm^2^/s) in air, low spark ignition energy (0.02 mJ), high combustion heat (285.8 kJ/mol), easy leakage and wide flammable range (4.75%), which makes it a potentially hazardous gas [1,4,5]. As hydrogen production and hydrogen fuel cell technologies develop rapidly, the need for hydrogen sensors to safely process hydrogen at all stages of production, distribution, storage and utilization will continue to grow [1,6,7].

Optical hydrogen sensors based on the surface plasmon resonances (SPRs) of metallic nanostructures have been widely studied [8,9,10,11,12]. Compared with traditional electronic sensors, optical sensors usually have many advantages. The main advantage is that no sparks are introduced when operating near the sensing area. In addition, they are compact, relatively inexpensive, and immune to electromagnetic interference. It is also possible to spatially separate the readout and the sensing areas for these sensors, enabling their application in harsh environments [12,13].

Palladium and its alloys are commonly used for hydrogen detection. After palladium is exposed to hydrogen, the local hydrogen pressure in palladium increases, and palladium hydride (PdHx) is formed through the hydrogenation process [5,14,15,16,17,18,19]. The composition of palladium hydride, and thus its optical properties, depends directly on the surrounding hydrogen concentration. The conversion process from Pd to PdH_x_ leads to a change in dielectric function [20]. The hydrogen absorption of palladium can be manifested in many different optical parameters, including spectral extinction/transmission/reflection amplitude, peak/dip position and full-width at half-maximum (FWHM) of the peak/dip [16,17,18]. Based on this, various palladium plasmonic hydrogen sensors have been demonstrated [21,22,23,24,25]. For practical applications, it is desirable to be able to fabricate these sensors in a cost-effective and scalable fashion.

In this work, we used colloidal lithography (CL) to fabricate large-area palladium nanohole and nanotriangle arrays. These Pd nanostructures were fabricated using scalable nanofabrication techniques and thus large-area sample preparation is possible without using sophisticated lithographic instruments. We varied their structural parameters to systematically investigate their hydrogen sensing performance. At 3% H_2_ concentration, nanohole arrays with a period of 500 nm showed the biggest intensity change of 5.5% at resonance wavelengths. 

## 2. Results and Discussions

Colloidal lithography, a scalable and versatile nanofabrication technique, has been widely employed to fabricate a variety of plasmonic nanostructures for different sensing applications [26,27,28,29,30]. The Pd NHA was fabricated on quartz substrates with a multi-step process, which is illustrated in Figure 1a. First, a close-packed monolayer of polystyrene (PS) nanospheres was formed on the quartz substrate through the self-assembly of the nanospheres, as reported in our previous work [31] and described in detail in the Materials and Methods section. Then the nanospheres were subjected to O_2_ plasma treatment with reactive-ion etching (RIE) to reduce the sphere size to a desirable value. Finally, with the etched nanosphere arrays as masks, 20 nm Pd was deposited onto the substrate followed by the removal of the PS nanospheres with ultrasonication in ethanol, leaving ordered Pd nanohole arrays on the substrate. The inset in Figure 1c shows a scanning electron microscopy (SEM) image of the fabricated Pd NHA. The nanohole size as well as the array periodicity is quite uniform over a large area, suggesting the quality of the PS nanosphere monolayers is good. The diameter of the nanoholes can be accurately controlled by O_2_ plasma etching time and thus the resonance wavelengths of the NHA can be readily tuned. In addition, using spheres with different diameters leads to corresponding changes in the array period, providing additional tuning means for these nanohole arrays. 

Numerical simulations were performed on such Pd NHAs to evaluate the changes in their optical properties upon H_2_ absorption. The simulation results, as shown in Figure 1b, show that the transmittance increases dramatically at resonance dips while the broad resonance peak at 750 nm shifts to a longer wavelength. This is due to the change in the optical permittivity *ε* of Pd to PdH_x_. The permittivity of Pd can increase by 20% under the exposure of H_2_ [32,33,34,35]. The momentum matching condition $\sqrt{\frac{\varepsilon_{1}\varepsilon_{d}}{\varepsilon_{1}+\varepsilon_{d}}}$ in Equation (1) become larger, and *λ_P_ (i*, *j)* moves to a longer wavelength. Figure 1c shows the experimental transmission spectra of a fabricated Pd NHA with *D* = 360 nm and *P* = 500 nm in the N_2_ environment and with a 3% hydrogen concentration, which qualitatively agrees with the results in Figure 1b. From the simulation and experimental results, it can be seen that all of the spectra show two peaks and two dips; we denote the peak/dip wavelengths as *λ*_*P*1_, *λ*_*P*2_, *λ*_*D*1_ and *λ*_*D*2_, respectively. According to previous reports [35,36,37,38,39], as shown in Equations (1) and (2), *λ*_*P*1_, *λ*_*P*2_, *λ*_*D*1_ and *λ*_*D*2_ can be assigned to the (1,1) Pd/glass and (1,0) Pd/glass resonance peaks, (1,0) Pd/glass and (1,0) Pd/air Wood’s anomaly transmission minima, respectively [35,37].
(1)$$\lambda_{P}\left(i,j\right)=\frac{\sqrt{3}}{2}\frac{P}{\sqrt{i^{2}+ij+j^{2}}}\sqrt{\frac{\varepsilon_{1}\varepsilon_{d}}{\varepsilon_{1}+\varepsilon_{d}}}$$
(2)$$\lambda_{D}\left(i,j\right)=\frac{\sqrt{3}}{2}\frac{P}{\sqrt{i^{2}+ij+j^{2}}}\sqrt{\varepsilon_{d}}\;$$

As shown in Figure 1b,c, *λ*_*P*1_ is broad and shifts to a longer wavelength after hydrogen absorption, making it difficult to track its spectral positions. Therefore, we monitored the transmittance changes in the samples in this study. Three sets of Pd nanohole arrays, i.e., P500D400, P500D360 and P300D200, were prepared and used to study the effect of different periods and hole sizes. The samples were placed inside a custom-built gas chamber. Hydrogen gas concentration in the chamber was controlled by regulating the flow rate of a 3% N_2_-diluted H_2_ gas and a pure N_2_ via two mass flow controllers. The total flow rate of the gas mixture was fixed at 300 sccm during the test. After the H_2_ concentration was set, we continuously recorded the transmission spectrum of the Pd NHA with a time interval of 2 min.

Figure 2 shows the hydrogen sensing performance of the three groups of Pd NHAs. Figure 2(a2,a3) shows the absorption and desorption, respectively, of the transmission spectra of P500D360 NHA under different hydrogen concentrations in the wavelength range from visible to near-infrared. When the Pd NHA is exposed to H_2_ gas, the transmission spectrum exhibits dramatic changes and the magnitudes of these changes are strongly wavelength-dependent. Before hydrogen absorption, the transmittance varies between 37% and 52% in the wavelength range of measurement. After hydrogen absorption, the transmittance varies between 42% and 49% with a smaller fluctuation range of 7%. The change in transmittance before and after hydrogen absorption can be attributed to the fact that the absorption of H_2_ leads to a negative dispersion factor *ε*_1_/*ε*_2_ decrease (*ε*_1_ and *ε*_2_ are the real and imaginary parts of the dielectric constant of the Pd/PdH_x_ film, respectively), which weakens the plasmon response of the nanoholes. In particular, the transmission increases at λ_*P*1_, λ_*D*1_ and λ_*D*2_, but decreases at λ_*P*2_. As the hydrogen concentration decreases, the transmission spectrum of the Pd NHA is gradually restored to its original position.

As shown in Figure 2(a2,b2), when the hydrogen concentration increases from 0 to 3%, the spectral positions of λ_*P*2_ of P500D360 and P500D400 exhibit considerable red-shifts with a corresponding sensitivity greater than 200 nm at 3% H_2_. For sample P300D200 (Figure 2(c2)), the sensitivity in terms of the wavelength shift is about 100 nm at 3% H_2_. The wavelength shift is generated by the formation of Pd hydride upon exposed to H_2_, leading to a variation in the permittivity of the Pd hole array. According to Equation (1), we can derive the wavelength sensitivity to the metal permittivity as follows:(3)$$\frac{d\lambda_{P}\left(i,j\right)}{d\varepsilon_{d}}=\frac{\sqrt{3}}{4\varepsilon_{d}}\frac{P}{\sqrt{i^{2}+ij+j^{2}}}\sqrt{\frac{\varepsilon_{1}^{3}}{{\left(\varepsilon_{1}+\varepsilon_{d}\right)}^{3}}}$$

According to Equation (3), the wavelength sensitivity is directly proportional to the hole array period. Therefore, it is expected that a larger sensitivity can be obtained with bigger PS spheres. Indeed, it is observed that the sensitivity of P500D360 and P500D400 sensors is much higher than that of a P300D200 sensor. Both P500D360 and P500D400 show similar sensitivity, indicating that the hole diameter plays a minor role in this diameter range. In our experiments, as the position of λ_*P*2_ eventually moves out of the range of the spectrometer. It is noted that the shift in resonance wavelength is inevitably accompanied by intensity changes. We thus monitored the transmission changes in the NHAs in a different hydrogen environment. Table 1 shows the performance of other reported plasmonic hydrogen sensors. It can be found that the NHAs proposed in this study show comparable or better sensitivity performance while the hydrogen absorption and desorption times are longer. If the sensor response time is defined as the time needed for the sensor to reach 80% of its stable response, the sensor response time of the NHAs is estimated to be 16 min at 3% H_2_.

For better visualization, we have displayed the transmittance changes in the samples in Figure 3. Δ*T* is defined as the difference between the spectral transmittance under a given hydrogen concentration and the transmittance in a N_2_ environment. As shown in Figure 3(a1), Δ*T* exhibits the largest variation at resonance dip λ_*D*1_ = 460 nm when P500D360 is exposed to 3% H_2_. The evolution of Δ*T* during H_2_ absorption and desorption is displayed in Figure 3(a2) showing a hysteresis-like curve. After the Pd NHA stayed in N_2_ for 5 min, hydrogen was introduced and its concentration gradually increased, leading to changes in the spectral transmittance. From 8 min on, the transmittance change increases rapidly before becoming saturated at 5.5% after half an hour when H_2_ concentration reaches 3%. Afterwards, as the H_2_ concentration is reduced, hydrogen molecules are released from the palladium lattice into the air, and the transmittance of the sample begins to decrease quickly. After the hydrogen is desorbed for about 20 min, the spectral transmittance basically returns to the initial value. A similar hysteresis-like curve is also observed for transmittance changes at *λ*_*P*1_ with smaller Δ*T* values.

The spectral transmittance of P500D400 for H_2_ absorption and desorption shows similar phenomena as shown in Figure 2(b2,b3) and Figure 3(b1–b3), illustrating that the nanohole diameters have little influence on the sensing performance of the Pd NHAs, which can be an advantage from the point view of fabrication tolerance.

Smaller PS nanospheres were also used to fabricate more compact nanohole arrays. The transmission spectra of P300D200 for H_2_ absorption and desorption are shown in Figure 2(c2,c3). Different from P500D360 and P500D400, the spectra of P300D200 show two transmission peaks and one resonance dip. The position of the transmission dip at *λ*_*D*2_ ≈ 420 nm is (1,0) Pd/air Wood’s anomaly transmission minima according to Equation (2). The two transmission peaks at *λ*_*P*1_ ≈ 350 nm and *λ*_*P*2_ ≈ 650 nm can be attributed to the (1,1) Pd/glass and (1,0) Pd/glass resonance peaks, according to Equation (1). Considering the smaller period, it is likely that λ_*D*1_ shifts to a shorter wavelength in the UV range. Its response towards H_2_ seems weak as the spectral transmittance shows smaller changes compared to Pd NHAs with P = 500 nm. At *λ*_*P*1_, Δ*T* = 1.2% is observed when the H_2_ concentration is 3%. Its hysteresis-like curve is shown in Figure 3(c1–c3). Although the transmittance changes are small for P300D200, it is noted that the wavelength shift of *λ*_*P*1_ is large, indicating a good sensing method in this wavelength range.

Through a finite difference time domain (FDTD) simulation, the transmission spectra of Pd NHAs were obtained to explore the effect of H_2_ absorption. Figure 2(a4,b4,c4) shows the simulated transmission spectra of P500D360, P500D400 and P300D200, respectively. It can be seen that there is a good consistency between the experimental transmission spectra and the simulated ones of the Pd-PdH_x_ process of Pd NHAs. The overall spectral features are consistent with those obtained experimentally, with some differences that can be attributed to the real refractive index changes in such thin Pd films.

In addition to nanohole arrays, ordered triangular nanostructures can be also fabricated by colloidal lithography with PS nanosphere monolayers as deposition masks. We prepared two Pd nanotriangle arrays (Pd NTA): P500 and P300, in which P represents the period of the array (i.e., the diameter of the PS spheres). Pd thin films were directly deposited onto unetched, densely packed PS sphere monolayers. Subsequent ultrasonication in ethanol produces periodic Pd nanotriangles on the substrate. Compared with Pd NHAs, the nanotriangles show weak plasmon modes with a relatively small peak magnitude. A similar H_2_ absorption and desorption process was carried out with these two samples and the results are shown in Figure 4 with different hydrogen concentrations. In the wavelength range of the experiments, the transmittance of both NTA samples increases with the increasing H_2_ concentration. When the concentration reaches 3%, ~2% changes are observed at most of the wavelengths of P500 with little changes in the plasmon spectral positions. Similar trends are also observed for NTA P300.

For both P500 and P300, the largest changes occur at ~750 nm. We monitored the transmittance changes at these wavelengths and their evolution curves are shown in Figure 5. Similar to the Pd NHA samples, both Pd NTAs exhibit hysterisis-like curves with similar maximum transmittance changes of ~2.5%. It can be concluded from these two samples that the change in period has little influence on the maximum position of transmittance changes for these Pd nanotriangle arrays.

## 3. Conclusions

We have demonstrated the nanofabrication of large-area Pd nanostructures (nanoholes and nanotriangles) with colloidal lithography and investigated their hydrogen sensing performances. It is observed that Pd nanohole arrays exhibit better-defined plasmon mode features and provide better sensing performances than nanotriangle arrays. Nanohole arrays with different diameters and periods are prepared and their spectral transmittance changes with different H_2_ concentrations are monitored. It is found that the sensing performance of the nanohole arrays greatly depends on the array period. Thus, within the detection range of the measurement system, it is better to choose bigger PS spheres. Considering the ease of fabrication of these scalable Pd nanostructures, this study could provide a guide to designing such large-area plasmon hydrogen sensors.

## 4. Materials and Methods

### 4.1. Nanohole Fabrication

Sulfate-functionalized polystyrene spheres with nominal diameter of 500 nm were purchased from Polyscience Inc. and dispersed in a water/ethanol mixture (1:1 volume ratio). Quartz substrates were cleaned in a mixture of NH_4_OH/H_2_O_2_/H_2_O (1:1:5 volume ratio) at 80 °C for 15 min and blown dry with N_2_. The nanospheres formed a closely packed monolayer on the glass substrates by colloidal lithography, as reported previously [31]. Diameters of the PS spheres were reduced by O_2_ plasma with reactive ion etching (RIE). The following etching parameters were used in our fabrication process: the oxygen flow rate was 10 sccm, the pressure of the chamber was 376 mTorr, the etching power and etching time were 100 W, 26 min for P500D400; 100 W, 34 min for P500D360; and 50 W, 14 min for P300D200, respectively. These etched sphere monolayers were then used as a template for Pd deposition with e-beam evaporation. The substrates were coated with 20 nm of Pd under a constant deposition rate of 0.01 nm/s. PS nanospheres were removed with sonication in ethanol for 5 min.

### 4.2. Optical Characterization

The optical transmission of Pd NHAs and Pd NTAs with different H_2_ concentrations was characterized by a homemade setup. Transmission spectra of Pd NHAs and Pd NTAs were measured by a fiber-coupled spectrometer (Ocean Optics, 2000PRO). The chamber and samples were flushed more than 10 times with hydrogen-nitrogen cycles. All the measurements were performed at 25 °C.

### 4.3. Numerical Simulation

Finite domain time difference (FDTD) method was used to simulate the transmission spectra of the nanohole arrays. A rectangular unit cell for calculation was used, with a 2-dimesional periodic boundary condition in *x*-axis and *y*-axis, and a perfectly matched layer (PML) boundary condition applied on the *z*-axis of the simulation. The refractive index of the quartz substrate was fixed at 1.5, and the optical parameters of Pd and PdHx were extracted from Ref. [33].

## Figures and Tables

**Figure 1 molecules-27-06100-f001:**
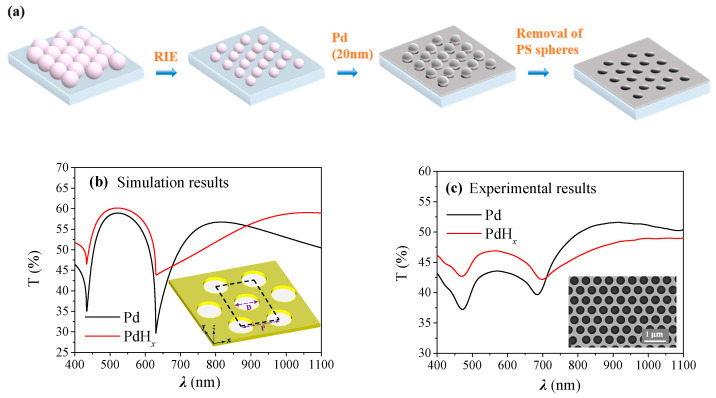
Fabrication and optical characterization of Pd NHA. (**a**) Schematic of the fabrication process of Pd NHA; (**b**) Simulated transmission spectra of Pd NHA (*D* = 360 nm, *P* = 500 nm) before (black) and after (red) H_2_ adsorption. The inset shows the geometrical configuration of Pd NHA and the unit cell for numerical simulations; (**c**) Experimentally measured transmission spectra of the Pd NHA (*P* = 500 nm, *D* = 360 nm) before (black) and after (red) H_2_ adsorption. The inset shows the SEM image of the nanohole arrays. *D* and *P* represent the diameter of the nanohole and the period of the array, respectively.

**Figure 2 molecules-27-06100-f002:**
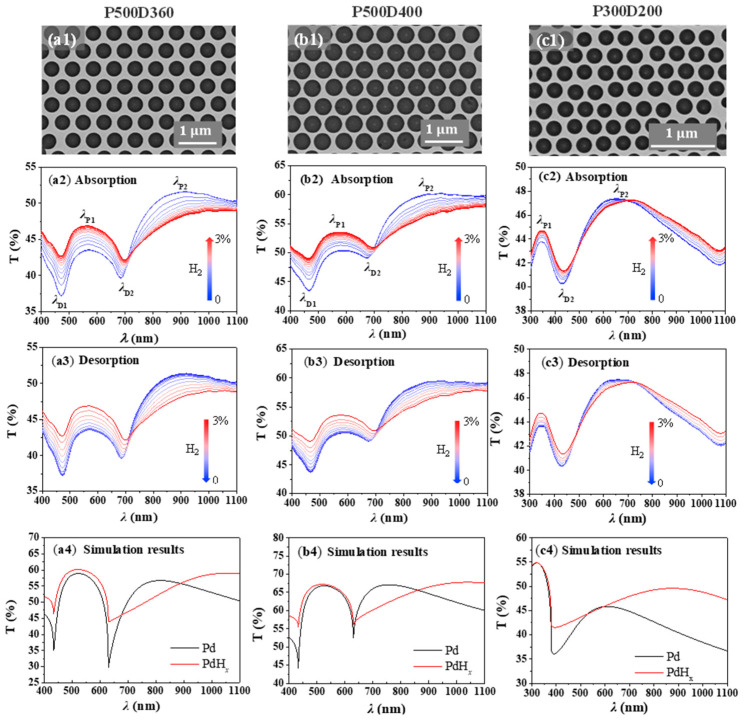
Optical transmission spectra and FDTD simulation results of Pd NHA samples upon hydrogen absorption and desorption processes: (**a1**–**a4**) P500D360; (**b1**–**b4**) P500D400; (**c1**–**c4**) P300D200.

**Figure 3 molecules-27-06100-f003:**
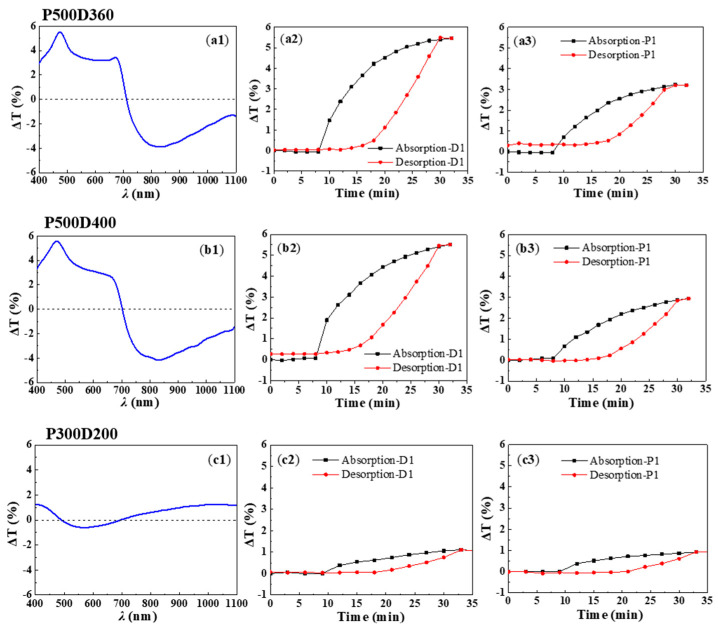
Spectral transmittance changes for different Pd NHAs: (**a1**–**a3**) P500D360; (**b1**–**b3**) P500D400; (**c1**–**c3**) P300D200.

**Figure 4 molecules-27-06100-f004:**
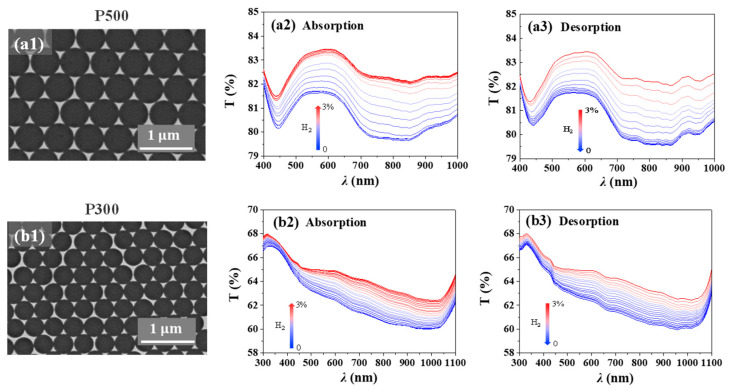
Optical transmission spectra of Pd NTAs during hydrogen absorption and desorption processes ((**a1**–**a3**) are the SEM morphology and transmission results of P500; (**b1**–**b3**) are the SEM morphology and transmission results of P300).

**Figure 5 molecules-27-06100-f005:**
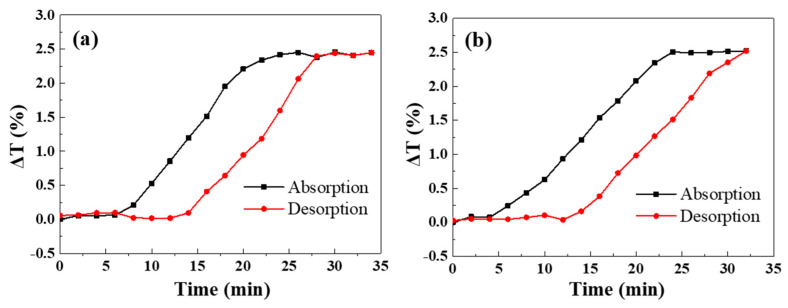
Transmittance change curve of Pd NTA samples upon hydrogen absorption and desorption processes ((**a**) P500; (**b**) P300).

**Table 1 molecules-27-06100-t001:** Performances of other reported hydrogen sensors.

Hydrogen Sensors Type	Sensitivity		Response Time	Recovery Time	Ref.
	Transmission	Wavelength			
Pd nano-disk	~6.5% at 3% H_2_	30 nm at 3% H_2_	<10 s	<30 s	[40]
Pd hole arrays	~7% at 2% H_2_	200 nm at 2% H_2_	-	-	[41]
Pd hole arrays	~7% at 2% H_2_	200 nm at 2% H_2_	-	-	[34]
Au nano-disk	~1.8% at 10% H_2_	~3 nm at 10% H_2_	0.3–1 s	-	[42]

## Data Availability

Data are available from the authors upon request.

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
