# Peer review of "Large-Area Ordered Palladium Nanostructures by Colloidal Lithography for Hydrogen Sensing"

_molecules, 2022, doi:10.3390/molecules27186100_

Round 1

Reviewer 1 Report

In the paper “Large-area ordered palladium nanostructures by colloidal lithography for hydrogen sensing” by Feng Xu et al., the authors demonstrated nanofabrication of large-area Pd nanostructures with colloidal lithography and investigated their hydrogen sensing performances. The work is solid. Therefore, I recommend its publication in molecules after the following comments are addressed:

1. Comparing the performance of the proposed H2 sensor with those of other H2 sensors, for example, the sensitivity, the absorption and desorption time, etc.

2. Provide more simulation or experimental results to give the optimal parameters of the Pd NHA for highest sensitivity. And explain why.

3. Modify the figures and the presentation to make the manuscript clearer, for example, Fig. 1b and 1c should be in the same row, the horizontal axis of Fig. 2 c2-c4 should be the same, Fig 4 a1-a3 should be in the same row, etc.

Reviewer 2 Report

The authors have fabricated large-area palladium nanohole and nanotriangle arrays by using the technique of colloidal lithography. The authors assert that their method is easier to implement when compared to other lithographic techniques in terms of their complexity. The authors experiment with different hole sizes and evaluate the performance of their hydrogen sensing. In general, it's an interesting piece of writing. A few nitpicks from me are as follows:

1) The results of the simulation are displayed in Figure 1b, and they demonstrate that the transmittance significantly increases at resonance dips, whereas the broad resonance peak that was originally located at 750 nm moves to a longer wavelength. Could the authors provide further explanation in reference ot these peaks?

2) Please provide citations for any equations that were taken from other published works.

3) There are some curves that are difficult to distinguish. To make things easier for your readers, you might want to use a different colour or symbols..

Despite the fact that the authors have put in a significant amount of effort into fabricating periodic arrays of nanoholes and running simulations, the conclusion is not very convincing and is presented in a confusing way. 
